# Plant-Based Diet and Glycemic Control in Type 2 Diabetes: Evidence from a Thai Health-Promoting Hospital

**DOI:** 10.3390/nu16050619

**Published:** 2024-02-23

**Authors:** Jonah Bawa Adokwe, Donrawee Waeyeng, Kanyamon Suwan, Kanchana Camsanit, Chanakan Kaiduong, Pawida Nuanrat, Phisit Pouyfung, Supabhorn Yimthiang, Jaruneth Petchoo, Soisungwan Satarug, Tanaporn Khamphaya

**Affiliations:** 1Environmental Safety Technology and Health, School of Public Health, Walailak University, Nakhon Si Thammarat 80160, Thailand; jonahbawa.ad@st.wu.ac.th (J.B.A.); ksupapor@mail.wu.ac.th (S.Y.); 2Office of Disease Prevention and Control Region 11, Nakhon Si Thammarat 80000, Thailand; donrawee.wae@gmail.com; 3Occupational Health and Safety, School of Public Health, Walailak University, Nakhon Si Thammarat 80160, Thailand; kanyamon.su@mail.wu.ac.th (K.S.); kanchana.ca@mail.wu.ac.th (K.C.); chanakankaiduong@gmail.com (C.K.); pawidanuanrat@gmail.com (P.N.); 4Department of Community Health, Faculty of Public Health, Mahidol University, Bangkok 20100, Thailand; phisit.pou@mahidol.edu; 5Department of Community Public Health, School of Public Health, Walailak University, Nakhon Si Thammarat 80160, Thailand; jaruneth.pe@wu.ac.th; 6Kidney Disease Research Collaborative, Translational Research Institute Woolloongabba, Brisbane, QLD 4102, Australia; sj.satarug@yahoo.com.au

**Keywords:** blood sugar, hyperglycemia, type 2 diabetes (T2DM), plant food score

## Abstract

The prevalence of type 2 diabetes (T2DM) is associated with diet. While consumption of plant-based foods may reduce blood sugar levels, the impact of consuming plant-based foods on fasting blood sugar levels has not been well defined. This cross-sectional study was conducted at the Health-Promoting Hospital in Pak Phun Municipality, Thailand. It included 61 patients with T2DM and 74 controls matched for age and gender. Dietary intake levels among T2DM and controls were assessed by a validated food-frequency questionnaire from which plant-based-food scores were calculated. This study found significant differences between specific plant foods and fasting blood sugar levels in patients with T2DM. Adherence to a plant-based diet appeared to influence fasting blood sugar levels. Patients who consumed higher amounts of certain vegetables and fruits showed lower fasting blood sugar levels. Diabetic patients consumed more legumes than controls, but the consumption of cereals and nuts/seeds in the two groups were similar. Consumption of nuts and seeds was also associated with a 76.3% reduction in the risk of a T2DM diagnosis. These findings suggest the potential efficacy of glycemic control in T2DM patients. More work is required to explore strategies for preventing and treating metabolic disorders through dietary modification.

## 1. Introduction

The global burden of type 2 diabetes mellitus (T2DM) continues to be a significant concern for public health nutrition. According to a 2021 estimate, at that time, 10.5% of the world population aged 20 years and older was living with T2DM [1]. According to the International Diabetes Federation, T2DM cases in Southeast Asia constitute 74% of 90 million individuals [2]. The 2014 NHES V in Thailand revealed diabetes prevalence rates of 9.9%, 8.9%, and 10.8% among total adults, males, and females [3]. Diabetes is a persistent medical condition marked by elevated blood sugar levels due to insulin resistance, reduced sensitivity of target cells to insulin, and disruptions of carbohydrate metabolism [4]. Dietary modification may help maintain blood sugar and HbA1c within normal levels of 100 mg/dL and less than 5.7%, respectively. Dietary modification may also prevent health conditions associated with hyperglycemia [5,6].

Evidence that plant-based diets may positively affect blood glucose levels comes from a study that linked a decreased risk of T2DM to a high intake of plant foods [7]. The simplest approach to exploring the quality of a plant-based diet involves using the plant food score (PFS). The PFS considers both the varieties and quantities of the plant food items consumed, which can include vegetables, fruits, legumes, nuts/seeds, and cereals [8,9]. Several studies have investigated the relationship between plant-based diets and T2DM [10,11]; however, the association between PFS and fasting blood sugar (FBS) levels has not been investigated thoroughly.

The primary objective of this investigation was to examine the correlation between PFS and FBS levels, specifically among residents of Pak Phun Municipality, Nakhon Si Thammarat Province, where the prevalence of T2DM is high [12]. The results of this study lend support to public health policies that focus on the mitigation and management of diabetes through the promotion of healthy dietary practices.

## 2. Materials and Methods

### 2.1. Participant Recruitment

This study employed a purposive sampling method to enlist subjects with T2DM and controls matched for age and gender (Ethic number: WUEC-21-223-01). Recruitment occurred at a local health center in Pak Phun Municipality, Nakhon Si Thammarat Province, Thailand, from June to December 2021. The inclusion criteria were as follows: participants must be residents of the Pak Phun municipality aged 50 or older who attended annual health checkups, who were diagnosed with T2DM, and who were apparently healthy. The exclusion criteria comprised non-resident status, current pregnancy and/or breastfeeding, and hospital records or a physician’s diagnosis of advanced chronic diseases including heart disease stroke, and cancer.

Participants were provided with the study objectives, study procedures, potential risks, and benefits, and they gave written informed consent before participation; sociodemographic data, educational attainment, occupation, health status, family history of diabetes, drinking and smoking status, medicines taken, and dietary supplement usage were obtained via structured interview questionnaires. After individuals with missing data were excluded, a total of 135 subjects (61 diabetics and 74 controls) were enrolled in this study.

### 2.2. Dietary Assessment

A validated Food Frequency Questionnaire (FFQ) was used to calculate Plant Food Scores (PFS) [13]. The questionnaire captured consumption frequency and typical portion size over a specified period.

### 2.3. Calculation of the Plant Food Score (PFS)

Inspired by work by Dennis and colleagues [14], we developed the below method for computing the PFS for the present study.

PFS values were assigned to five categories of plant foods: vegetables, fruits, legumes, nuts/seeds, and whole grains [14]. The category assignments are essential to enable assessment of the consumption of plant-derived chemicals [15]. Plant food samples of each category were weighed and adjusted to body weight (e.g., fruit g/kg b.w.). Intake levels were given scores of 0, 1, and 2 for low, medium, and high consumption, respectively, according to quartile ranking; <Q1 = 0, Q1–Q3 = 1, and >Q3 = 2.

The category PFS values for vegetables, fruits, legumes, nuts/seeds, and whole grains were calculated from the summation of individual plant scores in each category. The points from the five categories were summed to obtain the total PFS, where higher total scores indicated higher consumption rates of plant foods. Finally, the total PFS was grouped into low, medium, and high consumption by quartile ranking, as described above.

### 2.4. Plasma Glucose Determination

Participants were instructed to fast overnight, and their blood was drawn the next morning at the local health center in Pak Phun Municipality. Blood samples were collected in tubes containing sodium fluoride, a glycolysis inhibitor, for the glucose assay. These samples were kept on ice and transported within 1 h to the medical technology laboratory at Walailak University. Plasma glucose levels were determined by the glucose oxidase-peroxidase method (Glu Colorimetric Assay Kit, Elabscience, Houston, TX, USA) [16]. Fasting blood glucose levels of 70–99, 100–125, and ≥126 were considered suggestive of normal, prediabetes and diabetes, respectively.

### 2.5. Anthropometric Measurement

Height was determined with a portable stadiometer. A Tanita SC-330 analyzer (Tanita, Arlington Heights, IL, USA) was used for weight and body composition assessments, employing tetrapolar bioelectrical impedance analysis (BIA). Waist circumference (WC) measurements were taken using non-elastic tape. To ensure accuracy, the same team collected all data and instruments were calibrated daily.

### 2.6. Statistical Analysis

The data were analyzed using IBM SPSS Statistics 21 (IBM Inc., New York, NY, USA) and GraphPad Prism version 10 software (GraphPad, La Jolla, CA, USA). To assess the data distribution, a Kolmogorov-Smirnov test was conducted; the test results indicated that the data conformed to a normal distribution. Parametric statistical tests were then applied. Mean differences for continuous data were compared using an unpaired Student’s *t*-test for two groups or one-way ANOVA for three groups or more. The differences in percentage were assessed by a Pearson chi-square test. Logistic regression analysis was used to determine the odds ratio for high fasting blood sugar and a diagnosis of T2DM. A significance level of *p* ≤ 0.05 was set for all statistical tests.

## 3. Results

### 3.1. Characteristics of Controls and Diabetics

Participants in this study were those who underwent screening for T2DM, with consideration given to various demographic factors. No significant differences in education, occupation, income, or physical activity were observed between non-diabetic controls and diabetic patients (Appendix A).

As the data in Table 1 indicate, mean age and mean body mass index (BMI) were similar between control participants and participants with diabetes. The distributions of controls and participants with diabetes across BMI categories did not differ. Mean waist circumference was significantly greater in the T2DM group. This finding aligns with body fat data, with a higher percentage of “very high” body fat in participants with diabetes than in controls. Hypertension was more prevalent in the participants with diabetes than in the controls (60.7% vs. 41.9%). Mean fasting blood sugar was higher in the participants with diabetes than in the controls (177 vs. 94 mg/dL).

### 3.2. Comparing PFS in Diabetics and Controls

We used PFS to quantify the consumption of plant food groups. Quantities of vegetables, fruits, legumes, nuts/seeds, and cereals were adjusted for body weight. Consumption levels were categorized as low, medium, or high based on quartile values, contributing to the comprehensive Plant Food Score (PFS). Higher PFS scores indicated a higher consumption of plant foods. The total plant food scores revealed differences in the amounts of certain plant foods consumed among participants with diabetes who had different fasting blood sugar levels.

The mean consumption scores for total vegetables (holy basil, morning glory, and cassia) and fruit (pomelo, pineapple, and guava) were higher in the controls than in the participants with diabetes (Figure 1A,B). Participants with diabetes consumed more legumes but similar amounts of nuts/seeds and cereals compared to controls (Figure 1C–E).

Intake of plant foods such as fruits, nuts/seeds, and cereals, but not legumes, differed among subjects grouped by fasting blood sugar (FBS) levels (Figure 2).

### 3.3. Effects of Specific Plant Food Groups on FBS Levels

To identify specific plant foods that may affect FBS levels, we investigated the level of consumption of various plant foods in five categories. The results are shown in Figure 3.

We observed lower FBS levels in the group with higher intake levels of several plants, namely, morning glory, ivy gourd, Senna siamea, cowa leaves, edible fern, holy basil, and cashew leaves (Figure 3A). High consumption of soybeans, black beans, red beans, and green beans also resulted in lower FBS levels (Figure 3C). Similarly, lower FBS levels were found in those who consumed greater amounts of fruit, including pineapple, guava, pomelo, and pomegranate fruits (Figure 3B). Conversely, higher FBS levels were found in those with high consumption of long beans, winged beans, and acacia.

We had very limited data on nuts/seeds because these plant foods were not popular in this area. Nevertheless, data showed that a higher intake of cashew nuts resulted in lower FBS levels (Figure 3D). For the cereals group, lower FBS was found to correlate with high consumption of black sesame, basil seed, and corn (Figure 3E).

Collectively, these results suggest that higher consumption of plant food, particularly vegetables, fruits, and some legumes, produced lower FBS levels.

### 3.4. Associations of High FSB and T2DM Diagnosis with Plant Foods

To further evaluate the effect of plant foods on FBS, a logistic regression analysis was employed. The results show that consumption of legumes at a medium level was associated with a 3.72-fold increase in the likelihood of having FBS ≥ 100 mg/dL, compared to consumption of legumes at a low level (Table 2).

In another logistic regression analysis of 74 controls and 61 T2DM cases, consumption of nuts or seeds was associated with a 76.3% reduction in the risk of diagnosis with T2DM (Table 3).

## 4. Discussion

Utilizing PFS, this study investigated the relationship between dietary patterns and fasting blood sugar levels in individuals diagnosed with T2DM and controls matched for age and gender. We observed that there was a 76.3% reduction in the risk of diagnosis with T2DM in those who consumed nuts or seeds (Table 3). Lower fasting blood sugar levels were found within the group with higher intake of plant foods, especially fruits, nuts/seeds, and cereals (Figure 3). These findings suggest that consumption of specific plant foods may contribute to better management of fasting blood sugar levels among individuals with T2DM who reside in the southern region of Thailand.

The effect of the consumption of plant foods on fasting blood sugar levels was clear (Figure 2 and Figure 3). The association between lower fasting blood sugar and higher PFS scores indicates the potential benefits of a plant-rich diet. The results of this study are consistent with those reported in several publications from diverse regions, which collectively suggest that plant-based diets, particularly those rich in high-quality plant foods, are associated with a decreased risk of developing type 2 diabetes. For instance, (i) the Nurses’ Health Study (NHS) conducted across the US found that consuming plant-based diets, particularly those abundant in high-quality plant foods, is strongly linked to a considerable decrease in the risk of developing type 2 diabetes [17]; (ii) the Women’s Health Initiative (WHI) Clinical Trials in the USA found that greater adherence to a plant-predominant diet is associated with a lower risk of developing type 2 diabetes in post-menopausal women [18]; (iii) the Singapore Chinese Health Study showed that adherence to a high-quality plant-based diet is associated with lower risk of developing type 2 diabetes [19]; and most recently (iv), the Korean Genome and Epidemiology Study showed that the quality of plant foods may be important for the prevention of type 2 diabetes in the Korean population, who habitually consume diets rich in plant foods [20]. These findings underscore the potential global significance of plant foods in diabetes prevention and in transcending cultural and geographical boundaries.

The analysis of individual plant foods revealed a nuanced association with fasting blood sugar levels. High intake of certain plants, such as morning glory, ivy gourd, Senna siame, cowa leaves, edible fern, holy basil, cashew leaves, and *Glochidion p*., may have a lowering effect on blood sugar levels. In line with our findings, Adebayo F.A. et al. detailed the nutritional content of these plants [21]. The key mechanisms that may lead to lower blood sugar are their (i) high fiber content and/or (ii) polyphenols and antioxidants. Fiber consumption is known to prevent blood-sugar spikes by slowing digestion and reducing the rate of glucose absorption into the bloodstream. Polyphenols and antioxidants (found in plants like holy basil, *Ocimum sanctum*, and ivy gourd, *Coccinia grandis*, contain antioxidants and anti-inflammatory compounds [22,23,24] that combat oxidative stress, which is known to contribute to insulin resistance and worsen glycemic control. Polyphenols might also interfere with glucose uptake or carbohydrate catabolism [25,26]. The antioxidant properties of vegetables and fruits derived from plant-based diets have been documented previously [27]. These antioxidants, including vitamins C and E, beta-carotene, and various phytochemicals, protect against T2DM and mitigate the microvascular and macrovascular complications of T2DM. Other notable mechanisms involve (iii) micronutrients (vegetables are sources of minerals like magnesium and chromium, which are important in glucose metabolism and insulin function [28]), and (iv) low glycemic index (most leafy greens and non-starchy vegetables have a low glycemic index [29]). Taken collectively, this evidence suggests that plants rich in sources of these beneficial compounds may play a role in promoting metabolic health and reducing the risk of T2DM and its complications.

We observed higher legume scores among T2DM cases; in contrast, we found no differences in nut/seed and cereal scores compared with controls (Figure 1C–E). This finding diverges from previous research indicating that individuals in the highest quartile of total legume and lentil consumption had a lower risk of diabetes than those in the lowest quartile [30]. In addition, Yu and his team from Chongqing, China, reported a strong association between higher levels of consumption of legumes, nuts, and cereals and the following health outcomes: lower blood pressure, a reduced prevalence of hypertension, and improved blood-pressure control [31]. The discrepancy in our findings may be attributable to the complex nature of dietary patterns and their varied effects on health outcomes across different populations. Also in contrast to our results, a prospective cohort study suggested that the Japanese dietary pattern, which includes a higher intake of seaweeds, legumes, nuts, and mushrooms, could lower the risk of chronic diseases [32]. However, it is essential to acknowledge the potential methodological variations and sociodemographic factors that may contribute to divergent outcomes in different studies. A plausible explanation for our observation could be behavioral changes among patients with diabetes, driven by the need for better glycemic control. It is conceivable that individuals with diabetes may be inclined towards higher legume consumption to enhance their intake of protein and dietary fiber, aligning with the dietary recommendations for better blood sugar management. The intricate interplay between dietary choices, glycemic control, and the potential influence of cultural and regional dietary patterns underscores the complexity of understanding the relationship between legume consumption and T2DM. Further research is warranted to explore these dynamics and to elucidate the mechanisms underlying the observed associations.

Our study has strengths. First, it is situated in a region characterized by distinct dietary habits, offering valuable insights into the health effects of foods within the unique context of the Pak Phun community. Second, our dataset provides a comprehensive understanding of the specific dietary patterns prevalent in the community. Third, our findings provide individuals with T2DM with a practical dietary target for glycemic control: approximately 2.4 and 3.3 g/kg b.w./day for vegetable and fruit intake, respectively (Appendix A). To make these targets actionable, an adult weighing 60 kg aiming for glycemic benefits should strive for roughly two servings of various vegetables (144 g) and 1.3 servings of various fruits (198 g) daily. Individual adjustments may be necessary.

Of note, increased synthesis of bilirubin, an endogenous antioxidant, through activation of heme oxygenase-1 (HO-1) expression, could explain the observed protection against increased blood sugar associated with consumption of plant foods in our study. Consumption of green tea has been shown to increase HO-1 expression [33,34,35]. In a human trial that included only non-smoking diabetic subjects with no history of metabolic complications who did not take regular food supplements, green tea consumed in normal amounts increased HO-1 expression and reduced damage to the DNA in circulating lymphocytes [35]. A wide range of chemicals from plant foods, such as curcumin, catechin (in green tea), α-lipoic acid (in broccoli, and spinach), and sulforaphane (in cruciferous vegetables) are HO-inducers [36]. Thus, the beneficial health effects of consumption of these plant chemicals are, at least in part, mediated through increased HO-1 expression, which results in bilirubin synthesis and thus ROS neutralization.

We also acknowledge the limitations of our study. One significant methodological challenge was collecting and matching data for various age groups and ensuring adequate representation of female subjects. This limitation may be attributed to the experiment being conducted in one area only; therefore, further nationwide data collection should be considered. A lack of detailed information regarding specific dietary supplements and medications was another limitation of the present study, as was the short duration of data collection. An extended timeframe may yield more detailed insights. Moreover, certain plant categories lacked sufficient statistical power due to regional dietary habits, emphasizing the necessity for future research to address these gaps and investigate the broader implications for plant food consumption with regard to overall health.

## 5. Conclusions

This study observed a relationship between plant food scores and fasting blood sugar levels. Higher plant food scores from consumption of vegetables, fruits, nuts/seeds, and cereals were associated with lower fasting blood sugar levels. More work is required to explore plant-based diets that support healthy blood sugar levels, particularly when specific beneficial plant foods are used in Southern Thai cuisine. This information may aid physicians and dietitians in designing tailored nutritional interventions.

## Figures and Tables

**Figure 1 nutrients-16-00619-f001:**
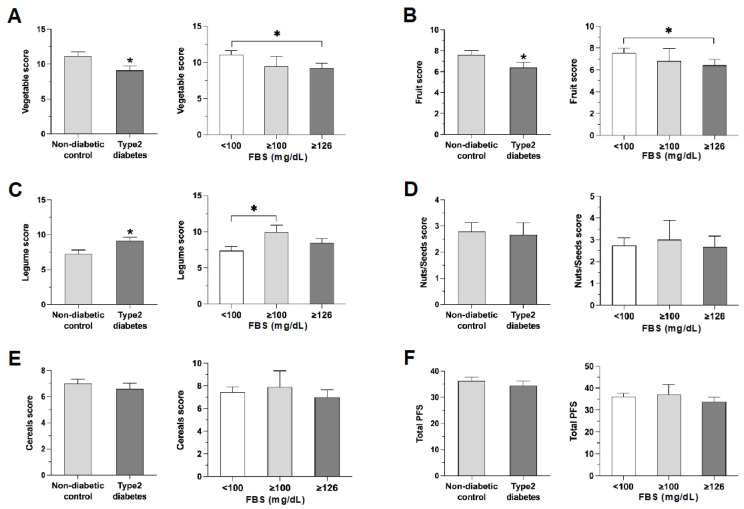
Comparing plant food scores in controls, diabetics, and groups with different FBS levels. The bar graphs represent (**A**) vegetable score, (**B**) fruit score, (**C**) legumes score, (**D**) nuts/seeds score, (**E**) cereals score, and (**F**) total PFS. Data represent the mean ± SEM. * *p* < 0.05, compared to the non-diabetic control or FBS < 100. Controls had higher vegetable and fruit scores, whereas participants with T2DM had higher legumes scores.

**Figure 2 nutrients-16-00619-f002:**
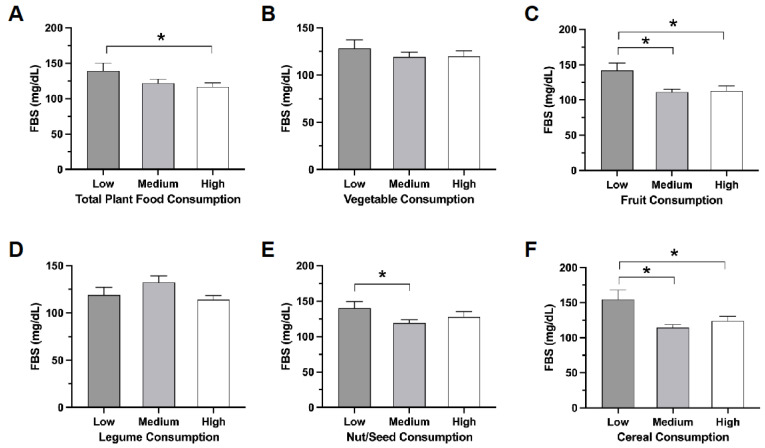
Comparing fasting blood sugar levels in participants grouped by plant food consumption. The bar graphs represent mean FBS values in participants in the lowest (low), medium (medium), and highest (high) score categories for (**A**) total plant food, (**B**) vegetable, (**C**) fruit, (**D**) legumes, (**E**) nuts/seeds, and (**F**) cereals consumption, with FBS levels on the y-axis. Data represent the mean ± SEM. * *p* < 0.05. Mean FBS was lower in participants in the high-consumption groups compared to those in the medium or the low-consumption group for fruit, nuts/seeds, and cereals, but not for legumes.

**Figure 3 nutrients-16-00619-f003:**
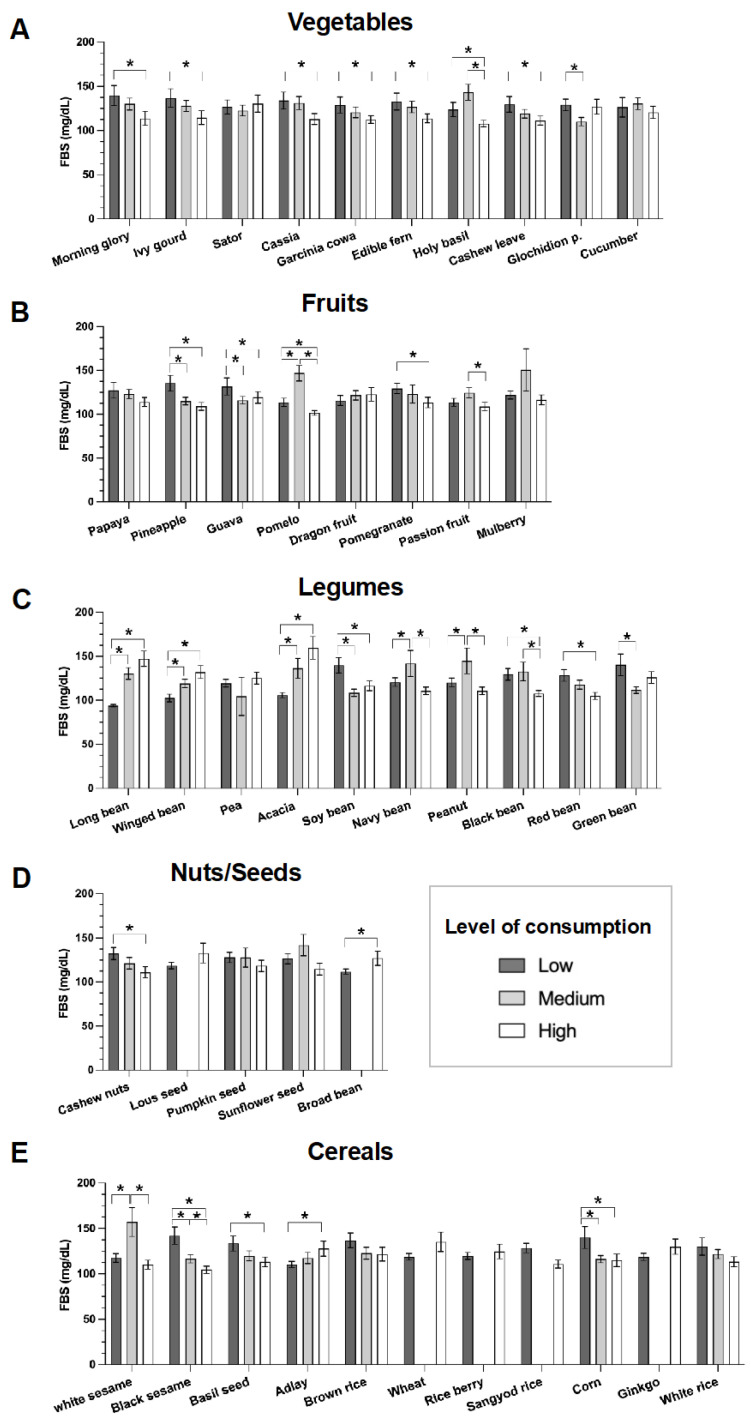
Effects of plant food consumption on fasting blood sugar levels. (**A**) vegetables (**B**) fruits (**C**) legumes, (**D**) nuts/seeds, and (**E**) cereals. The consumption level of each plant food was described as low, medium, or high, as defined in Section 2.3. Data represent mean FBS ± SEM. * *p* < 0.05.

**Table 1 nutrients-16-00619-t001:** Descriptive characteristics of non-diabetic controls and patients with type 2 diabetes.

Parameters	Non-Diabetic Control, n = 74	Participants with Type 2 Diabetes, n = 61	*p*-Value
**Age, years**	60.10 ± 1.081	59.90 ± 1.320	0.089
(Min, Max)	(41, 80)	(41, 81)	
**FBS, mg/dL**	93.66 ± 1.243	177 ± 9.354	0.001
(Min, Max)	(74, 143)	(110, 509)	
**Female, n (%)**	60 (81.10)	48 (78.7)	0.729
**BMI, kg/m^2^**	24.75 ± 0.545	25.88 ± 0.646	0.087
(Min, Max)	(14.69, 36.04)	(15.09, 48.13)	
Underweight (<18.5)	3 (4.05)	6 (9.84)	
Normal weight (18.5–22.9)	24 (32.43)	12 (16.67)	
Overweight (23.0–24.9)	12 (16.22)	8 (13.11)	
Obese (25.0–29.9)	30 (40.54)	25 (40.98)	
Extremely obese (>30)	5 (6.76)	10 (16.39)	
**Waist circumference, (cm)**	88.28 ± 1.267	93.60 ± 1.468	0.003
(Min, Max)	(145, 174)	(65, 176)	
**Smoker, n (%)**	9 (12.2)	7 (11.5)	0.902
**Alcoholic n (%)**	1 (1.40)	5 (8.20)	0.055
**Medicine n (%)**	6 (8.11)	48 (78.69)	0.000
**Dietary supplements n (%)**	3 (4.05)	7 (11.48)	0.101
**Hypertension, n (%)**	31(41.9)	37 (60.7)	0.030
**SBP, mmHg ≥ 140**	134.30 ± 1.851	142.10 ± 2.182	0.004
(Min. Max)	(141, 173)	(158, 183)	
**DBP, mmHg ≥ 90**	83.23 ± 1.084	84.42 ± 1.265	0.234
(Min, Max)	(91, 103)	(91, 107)	
**Body composition, body fat, n (%)**			
Low	33 (44.6)	27 (44.3)	0.016
Normal	24 (32.4)	9 (14.8)	
High	13 (17.6)	13 (21.3)	
Very high	4 (5.40)	12 (19.7)	

Values for age, FBS, BMI, waist circumference, SBP, and DBP are presented as the mean ± SEM. Sex, BMI classes, smoking status, alcohol consumption, and body fat percentage were reported as percentages. SBP: systolic blood pressure; DBP: diastolic blood pressure, with hypertension defined as SBP ≥ 140 mmHg or DBP ≥ 90 mmHg. A significance level of *p* ≤ 0.05 was used to identify statistical significance, as determined with the Pearson Chi-Square test for % differences and Student’s *t*-test for the means of two groups.

**Table 2 nutrients-16-00619-t002:** Logistic regression analysis for association of consumption of plant foods and risk of high fasting blood sugar ^a^.

Parameters	FBS (<100 mg/dL) n = 55 (%)	FBS (≥100 mg/dL) n = 80 (%)	Adjusted OR ^b^	95%CI	*p*-Value
Total PFS					
Low	11 (20.00)	22 (27.50)	Ref		
Medium	27 (49.09)	34 (42.50)	0.984	0.322–3.013	0.978
High	17 (30.91)	24 (30.00)	0.992	0.296–3.324	0.990
Vegetables					
Low	14 (25.45)	21 (26.25)	Ref		
Medium	29 (52.73)	39 (48.75)	1.192	0.391–3.631	0.758
High	12 (21.82)	20 (25.00)	1.898	0.520–6.926	0.332
Fruits					
Low	14 (25.45)	20 (25.00)	Ref		
Medium	27 (40.09)	41 (51.25)	1.733	0.566–5.308	0.336
High	14 (25.45)	19 (23.75)	3.121	0.834–11.678	0.091
Legumes					
Low	21 (38.18)	15 (18.75)	Ref		
Medium	22 (40.00)	47 (58.75)	3.723	1.255–11.045	0.018 *
High	12 (21.82)	18 (22.50)	0.822	0.198–3.408	0.787
Nuts/seeds					
Low	20 (36.36)	32 (40.00)	Ref		
Medium	24 (43.64)	25 (31.25)	1.056	0.385–2.894	0.916
High	11 (20.00)	23 (28.75)	1.215	0.359–4.109	0.754
Cereals					
Low	12 (21.82)	26 (32.50)	Ref		
Medium	32 (58.18)	29 (36.25)	0.751	0.251–2.249	0.609
High	11 (20.00)	25 (31.25)	1.183	0.352–3.977	0.786

^a^ High fasting blood sugar was defined as FBS ≥ 100 mg/dL. ^b^ Adjusted for sex, education, income, supplements, and medications. * *p* < 0.05.

**Table 3 nutrients-16-00619-t003:** Logistic regression analysis for association of consumption of plant food and risk of T2DM diagnosis.

Parameters	Controls n = 74 (%)	T2DM Cases, n = 61	Adjusted OR ^a^	95%CI	*p*-Value
Total PFS					
Low	13 (17.57)	20 (32.79)	Ref		
Medium	36 (48.65)	25 (40.98)	0.499	0.141–1.768	0.281
High	25 (33.78)	16 (26.23)	0.267	0.062–1.162	0.078
Vegetable					
Low	15 (20.27)	20 (32.79)	Ref		
Medium	40 (54.05)	28 (45.90)	0.420	0.116–1.521	0.186
High	19 (25.68)	13 (21.31)	0.236	0.093–1.794	0.236
Fruit					
Low	16 (21.62)	18 (29.51)	Ref		
Medium	35 (47.30)	33 (54.10)	1.137	0.332–3.891	0.838
High	23 (31.08)	10 (16.39)	0.768	0.166–3.553	0.736
Legume					
Low	26 (35.14)	10 (16.39)	Ref		
Medium	34 (45.95)	35 (57.38)	3.551	0.887–14.220	0.073
High	14 (18.92)	16 (26.23)	1.002	0.199–5.048	0.998
Nuts/seeds					
Low	23 (31.08)	29 (47.54)	Ref		
Medium	35 (47.30)	14 (22.95)	0.237	0.064–0.872	0.030 *
High	16 (21.62)	18 (29.51)	0.343	0.079–1.500	0.155
Cereals					
Low	15 (20.27)	23 (37.70)	Ref		
Medium	43 (58.11)	18 (29.51)	0.353	0.099–1.254	0.107
High	16 (21.62)	20 (32.79)	0.789	0.205–3.041	0.731

^a^ Adjusted for sex, education, income, supplement, and medicine. * *p* < 0.05.

## Data Availability

All data are included in this article.

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
