# Peer review of "Plant-Based Diet and Glycemic Control in Type 2 Diabetes: Evidence from a Thai Health-Promoting Hospital"

_nutrients, 2024, doi:10.3390/nu16050619_

Round 1

Reviewer 1 Report

Comments and Suggestions for Authors

The submitted manuscript addresses a significant and insufficiently researched topic: the correlation between a plant-based diet and glycaemic control in patients with type 2 diabetes. The study is a valuable contribution to the existing literature and meets the demand in this field. The study's results could serve as a valuable educational resource for dietitians and other health promotion practitioners. This is particularly relevant given the growing prevalence of carbohydrate disorders resulting from unhealthy dietary habits and abnormal lifestyle factors.

The article has a well-organized structure, following the typical layout for original papers and including all required sections. While some sections are well-developed, others, such as material and methods, results, discussion, and conclusion, may benefit from minor or major revisions.

The introduction provides sufficient background and includes all relevant references. All cited sources are relevant to the research. In the "Introduction" and "Discussion" sections, the literature data are well summarized.

The following are comments on specific parts of the manuscript:

Abstract

1) The abstract lacks an element of methodology.

Introduction

1) Line 39 – 10.5% including children and adolescents, or only in the population over 18 years of age?

2) Line 48 - < 110 mg/dl or 100 mg/dl? - normal fasting blood glucose is considered to be < 100 mg/dl. Please clarify.

3) Line 58 – fasting? blood glucose

Materials and Methods

1) Inclusion and exclusion criteria need to be clarified. Were only people without type 2 diabetes eligible for the control group? What about people with pre-diabetes (abnormal fasting glycaemia and/or impaired glucose tolerance)? What exactly does 'in good health' mean? This needs to be clarified. What chronic diseases were included in the exclusion from the study? (lines 70-72).

2) No information is available on whether the interviewers asked questions about the medicines and dietary supplements taken and the treatment of diabetes.

3) Was a fasting blood glucose ≥126 mg/dl the only criterion for the diagnosis of diabetes? A single fasting blood glucose result ≥126 mg/dl is not the basis for the diagnosis of diabetes - this result must be found in two measurements, each on a different day. What about a 120-minute OGTT result ≥ 200 mg/dl and HbAc1?

4) The way in which the PFS was estimated (in total and by the five categories) needs to be completed.

5) What method was used to determine blood glucose levels. Was it an enzymatic spectrophotometric method with glucose peroxidase?

6) The methodology for measuring glycaemia has been provided, however, the method of glycaemia classification has not been explicitly stated. This information can only be found in the results section, under the tables. It would be appreciated if additional details on these analyses could be provided. Furthermore, there is a lack of information provided on anthropometric measurements, body composition, and blood pressure. To further enhance the text, it is suggested that a new subsection 2.5 be added for 'Other Analyses' and a new subsection 2.6. be added for 'Statistical Analyses'.

7) Subsection 2.5. could benefit from further clarification regarding the normality of the data distribution and the homogeneity of variance.  It would be helpful to include information on which tests were conducted to address these issues.

Results

1) In describing the results, it may be beneficial to consider presenting the rounded values in brackets instead of repeating the exact mean and p-values included in the tables. Furthermore, it is crucial to specify the direction of the results (similar, higher or lower) for each group. Additionally, it could be valuable to provide information on the proportion of control and T2DM subjects who were normal weight, overweight, obese and abdominal obesity, similar to body fat content.

2) Line 116 - what does "diabetes level" mean?

3) Lines 122-123 - the sentence is unclear and requires rewording

4) Lines 143-152 - some of the explanations below the table refer to parameters that are not in the table. Some of this information should be moved to section 2.

5) Table 1 requires a review of the percentages in brackets for body composition. It is recommended that these discrepancies be addressed and corrected as necessary. It appears that some of the results may be incorrect. For instance, it seems that the percentage for 51 out of 74 subjects in the control group should be 68.9% instead of 44.6%.

6) Lines 154-159 - this fragment of the test fits the methodology better.

7) The same PFS results in the control and T2DM groups are presented in the table and figures. One way of presenting the results should be selected. It is worth supplementing the data with PFS results depending on FBS divided into groups (control and T2DM).

8) Lines 203-207 summarize the results. This part should be moved to the discussion.

9) In the title of subsection 3.3. the term "Association with Chronic Diseases" is exaggerated because the link only relates to glycemic control.

Discussion

1) The first paragraph of the Discussion should deal with a brief presentation of the results obtained in the study - please complete this.

2) The discussion is too long and needs to be shortened. Authors should decide which results they want to discuss.

3) Greater emphasis should be placed on explaining the mechanisms of the impact of ingredients present in plant products on glycemic control.

Conclusions

1) Line 34 - this is not a conclusion from the study, because the risk of diabetes was not assessed.

Other comments include:

·         Type 2 or type II DM - should be standardized in all parts of the manuscript;

·         T2DM - the abbreviation should also be entered in the abstract (line 19) and used in line 20;

·         All abbreviations should be explained in the first place of their introduction and then used without re-explaining. Multiple repetitions apply to the following abbreviations: PFS, FBS, T2DM;

·         Line 43 – between “[2].” and “In …” - space is missing;

·        Line 59 - not explained what is meant by Pb-related T2DM

·        Line 123 - unnecessary space between "44.30" and "%"

·        References to figures and tables are unnecessary in the discussion.

Overall assessment of the manuscript

Overall, the article is well-written and easy to understand. The Authors achieved their goals, which allowed them to draw conclusions, and accurately identified some limitations and strengths of the study. It is worth adding the public health implications of the findings. I recommend the manuscript for publication after appropriate revisions.

Reviewer 2 Report

Comments and Suggestions for Authors

Diabetes is a disease that affects a large part of the population. In the manuscript entitled "Plant-Based Diet and Glycemic Control in Type 2 Diabetes: Evidence from a Thai Health Promoting Hospital", the authors examined the correlation between plant-based diet and DM2 at a local health centre in Baiboon City. It is part of Nakhon Sitamara Province in Thailand.

This hypothesis is supported by a study that suggests that a plant-based diet may have a positive effect on blood sugar control, as higher intake of plant-based foods is associated with a lower risk of type 2 diabetes.

There are some key points to this study that need to be explored further.

1. One limitation of this study is the diagnosis of diabetes based on fasting blood glucose levels. 126 mg/dl). Is only this variable used? Recommendation: Use the HBA1c value.

2. There is a lack of better descriptions of people with diabetes. Do they offer any treatment? What drugs do they use? How long will it take?

Do they use insulin?

3. In my opinion, this data alone is not enough to draw conclusions from a wide variety of plants. In addition, the study did not consider any mechanism of action.

Reviewer 3 Report

Comments and Suggestions for Authors

What is most problematic with this manuscript are the descriptions of and methods used to calculate the plant food score (PFS). The authors claim that they adopted the methods of Dennis et al., 2021, but the actual similarities to those methods are minor.

(1) Dennis et al. (2021) used the validated Block FFQ to collect dietary intake data and valid methodology to separate complex dishes into component quantities of food. In this manuscript, the authors did not describe their FFQ nor indicate if it were validated for use in Thailand. Additionally, the authors reported that they used a structured 24-hr recall but did not describe what they meant by “structured” or how they intended it to capture the usual food intake of their participants based on a single recall.

(2) Dennis et al. (2021) reported their various categories of PFS as g/kcal intake. However, in the current manuscript, the various plant food scores are reported as mg/kg BW/day. The values do not make sense. For example, the mean PFS for “vegetable” for a non-diabetic control as 11 mg/kg/day. Assuming a BW of 70 kg, this comes out to 770 mg of vegetable per day or less than 1 gram. It also does not make sense if this was a typo and the values are in g/kg/day. The “vegetable” score is higher than the “cereal” score in a rice eating country! The “legume” score is close to the “cereal” score when the authors themselves reported that very few legumes are consumed in Thailand. Are some of these PFS based on cooked whereas others on raw foods?

With regards to the statistical analysis, the authors claim that they matched participants in the two groups (non-diabetic, diabetic) based on some variables. However, dietary habits are greatly impacted by educational level, income and sex. The statistical tests were not adjusted for education or income. The higher amount of “vegetable” (as an example) consumed by the non-diabetic group may have simply been because of income or education. (The supplementary table bears that out). The associations between PFS and blood sugar or blood pressure must be computed based on appropriately set-up statistical models and which are appropriately controlled.

The manuscript must be edited for both appropriate scientific writing and English writing. The errors are too numerous to mention. For example and starting with the Abstract:

Line 23 - It is incorrect to say that the study uncovered significant "associations"! The statistical tests did not examine associations. It only showed differences in consumption of some foods between the diabetic and non-diabetic group.

Line 24 - It is incorrect to say there were improvements in glycemic control and reductions in fasting blood sugar. This study was not a randomized controlled intervention study with an ability to show improvement. The only thing it showed were differences in the amount of the different plant food consumed or difference in PFS!

There are redundancies in data reported in tables and figures.

Not all data shown in tables needs to be repeated in the results section.

Much of the discussion is irrelevant. Must focus on topics related to the study.

The authors must understand that the statistical methods used in this study are mainly descriptive. If they want to report on associations, they need to run the appropriate statistical models. 

Comments on the Quality of English Language

This manuscript needs extensive editing for proper scientific language usage and English language usage. 

Reviewer 4 Report

Comments and Suggestions for Authors

The study aims to uncover the correlation between plant food consumption and blood glucose levels in patients with T2DM. The study was well-conducted, but the manuscript requires some modifications.

The major problem is that you made a conclusion that a high plant food score is associated with a lower risk of diabetes, which is common sense and not scientific. You need to define clearly how much consumption is “high” to let the T2DM patients know what the exact good plant food score for them is.

Some minor revisions also need to be made:

1.     Only need to write “Type 2 Diabetes Mellitus (T2DM)” once in the manuscript (L37), and write only “T2DM” in the following text. For example, L50, L54, L65. Same thing for “Plant Food Score (PFS)” and “fasting blood sugar (FBS)”.

2.     L40-L42. The sentence is too long. Check the grammar.

3.     L82. There should be a “.” after “over a specified period [12]”.

4.     L89. Define “low””medium” and “high” intakes.

5.     L108. Not “p-values” but “P values”.

6.     L115. Not “p = 0.089” but “P = 0.089”. Same thing for the rest.

7.     L157. Explain how much consumption of each type of food you define as “low””medium” and “high”. Give the exact numbers.

8.     Figure 3. Re-align the food names on the x-axis. “Type of consumption” can be changed to “Level of consumption”.

Round 2

Reviewer 4 Report

Comments and Suggestions for Authors

it could be accepted